# Renamer: A Transformer Architecture Invariant to Variable Renaming

## Abstract

Many modeling tasks involve learning functions which are invariant to certain types of input transformations. In this work we consider a specific class of invariance: semantics-preserving variable renaming. We first show that Transformer networks trained on such tasks do not always mirror the invariance of the underlying function. In this work we propose Renamer, a Transformer architecture which is invariant to semantics-preserving variable renaming. Learning a neural surrogate of a large-scale CPU simulator using Renamer reduces simulation error by 24.79-52.8% compared to using a vanilla Transformer. Furthermore, the invariant network is not sensitive to variable renaming, and its output remains constant when evaluated on a variable renamed version of the test set. Finally, Renamer is more efficient to train: it matches the performance of the vanilla Transformer with 24.75-59.06% fewer training epochs.

## 1 Introduction

Modeling tasks often require reasoning about *invariances* of the task with respect to transformations on the inputs (Snavely, 2019; Bianchi et al., 2022). A common approach to learning models that are invariant to a given transformation is to train models using data augmentations that exhibit the transformation (and corresponding invariance) under study (Shorten and Khoshgoftaar, 2019; Feng et al., 2021). However, such approaches give no formal guarantees that the resulting models are always perfectly invariant to the transformation. Further, there is evidence that baking the inductive bias of the invariance into the model leads to accuracy improvements (LeCun and Bengio, 1995; Cohen and Welling, 2016; Lee et al., 2019; Keriven and Peyré, 2019; Wang et al., 2022).

**Renaming invariance.** We study *renaming invariance*, a particular type of invariance in sequence processing tasks which arises when reasoning about formal languages including programming languages (Chen et al., 2021; Alon et al., 2019; Renda et al., 2020), mathematics (Lample and Charton, 2020; Polu et al., 2022), and synthetic grammars of natural languages (Marzoev et al., 2020; Berant and Liang, 2014). Renaming invariance is invariance to bijective transformations of the input tokens that preserve the semantics of the input. An example of renaming invariance is in symbolic algebra programs, where variables can be bijectively renamed and the result of evaluating the expression doesn't change.

**Renaming sensitivity.** General-purpose neural network architectures like LSTMs (Hochreiter and Schmidhuber, 1997) and Transformers (Vaswani et al., 2017) have shown impressive results on learning functions with renaming invariance (Alon et al., 2019; Renda et al., 2021). However, these neural networks do not themselves demonstrate renaming invariance. For example, Alon et al. (2019) note sensitivity to "uninformative, obfuscated, or adversarial variable names". This sensitivity presents a challenge to deploying neural networks in this context, as their predictions are not robust to semantics-preserving input transformations.

**Our approach.** We present an approach to enforcing renaming invariance in Transformers. The first key contribution that enables our approach is a formal definition of renaming invariance.

We define renaming invariance as a property of functions which take sequences of tokens as input. We first define a *view mapping* as a mapping from an input token to its *view*, the

semantic information about the token that is salient to the function. We then define a *referent mapping* as a mapping from an input token to its *referent*, the underlying entity to which it refers. A renaming invariant function is a function which generates the same output for any bijection of tokens that does not change tokens' views and bijectively renames tokens' referents – that is, salient semantic properties of tokens must not change, pairs of tokens which originally referred to the same underlying entity must both refer to the same permuted underlying entity, and pairs of tokens which originally referred to different underlying entities must still refer to different permuted underlying entities.

We present two architecture changes that together enforce renaming invariance in Transformers. We refer to the resulting architecture as the Renamer.

**View anonymization.** The first change, *view anonymization*, effectively replaces each token with a token that describes only its view. This enforces that the network is renaming invariant, because the network cannot make different predictions for different views. However, view anonymization alone reduces the representational capacity of the network, since the network can no longer distinguish tokens with different referents.

**Referent binding.** To recover the representational capacity, we introduce a novel modification to the attention layer which we call the *referent binding*. Referent binding restricts the attention in the first layer of the Transformer, allowing tokens to only attend to other tokens with the same referent. This breaks the symmetry between tokens with the same view but different referents, restoring the representative capacity of Renamer, while maintaining that Renamer is renaming invariant.

**Contributions.** We present the following contributions:

- We introduce and formally characterize the renaming invariance problem.
- We propose the two-step process of view anonymization and referent binding to enforce renaming invariance while maintaining representational power. We implement these in the Renamer, a renaming invariant Transformer model architecture.
- We evaluate the Renamer on a renaming invariant x86 assembly processing task. Renamer reduces the error compared to a vanilla model by between 27.58% and 52.80%.

By identifying and defining renaming invariance and proposing a Transformer model invariant to renaming invariance, our work takes a key step towards the goal of providing high-accuracy models with provable guarantees for tasks with input invariances.

## 2 Renaming Invariance in llvm-mca

This section presents a case study of renaming invariance in a sequence processing task. We first introduce *x86 basic block throughput prediction* and describe how it is a renaming-invariant task. We describe the views and referents present in x86 basic blocks for this task. We then describe renaming invariant permutations for this task, and show that the task's labels are invariant to these permutations, but are not invariant to other permutations. We finally demonstrate that Renamer generates accurate and renaming invariant predictions for this task, while baseline models are not renaming invariant and are therefore less accurate.

```
mov  64(%rsp), %rax        mov  64(%r8),  %rax        mov  64(%rax), %rax
sub  $1,    56(%rbp)        sub  $1,    56(%rbp)        sub  $1,    56(%ebp)
mov  16(%rax), %eax        mov  16(%rax), %eax        mov  16(%rax), %eax
   llvm-mca: 1.68 cycles       llvm-mca: 1.68 cycles       llvm-mca: 10.03 cycles

     (a) Original block.         (b) Invariant renaming.     (c) Non-invariant renaming.
```

Figure 1: Example of an x86-64 basic block and invariant and non invariant renaming. The registers may be renamed, as long as each register is renamed consistently.

**Task under study.** Following Renda et al. (2021) our task is to create a neural network surrogate of llvm-mca, a CPU simulator included in the LLVM compiler infrastructure (Lattner and Adve, 2004). As input llvm-mca takes a *basic block* of x86-64 assembly language which is a sequence of assembly instructions with no jumps or loops. It then outputs a prediction of the *throughput* of the basic block on the simulated CPU which is a prediction of the number of CPU clock cycles taken to execute the block when repeated for a fixed number of iterations. Learning a surrogate of llvm-mca results in faster or more accurate throughput estimation than using llvm-mca itself (Renda et al., 2021).

**Input specification.** We evaluate using a dataset of AT&T-syntax x86-64 basic blocks (Chen et al., 2019). Figure 1 presents three such basic blocks. AT&T syntax basic blocks are sequences of instructions, where each instruction consists of an opcode (e.g., `mov`), a source operand (e.g., `64(%rsp)`), and a destination operand (e.g., `%rax`). Each operand may be a constant (e.g., `$1`), a register (e.g., `%rax`), or a memory address (e.g., `64(%rsp)`).

Registers are the variables of x86-64 assembly. A given register operand consists of a *bitwidth* and a *base register*. The bitwidth is how many bits of the register data are addressed by the register. As an example, `%rax` addresses all 64 bits of the register data, `%eax` addresses the lowest 32 bits, and `%ax` addresses the lowest 16 bits. The base register is the location where register data is stored; for general-purpose registers this is indicated by the final several characters of the register (e.g., `ax` in `%rax`).

**Simulation model.** When executing an instruction, llvm-mca first waits for all of its source operands to be ready: a source operand is ready when all previous instructions that have matching destination operands (all predecessors in the *register dependency graph*) have finished executing. Once an instruction starts executing, its execution time is controlled by the bitwidth of each operand.

**Renaming invariance in llvm-mca.** Renaming invariance manifests in llvm-mca as invariance to register renaming. When the base register names are renamed in a given block such that none of the register bitwidths, nor the register dependency graph change, llvm-mca generates an identical prediction for this block, meaning that it is invariant to this class of transformation.

To formalize this, we say that the *view* of a given register is its bitwidth, and that the *referent* of a given register is the base register that it refers to. Any transformation of registers that maintains register views and is a bijection on register referents is a renaming invariant transformation.

**Example.** Figure 1 presents three x86 basic blocks in AT&T syntax. Figure 1(a) shows the original block. This basic block has a throughput of 1.6 cycles per iteration in llvm-mca's model. There are four unique registers in the basic block: `%rsp`, *%eax*, *%rax*, and **%rbp**.

The registers `%rsp`, *%rax*, and **%rbp** all share the same view as they are all 64 bit registers. *%eax* is a 32 bit register, and thus does not share the 64 bit width view but instead has its own view.

The registers *%eax* and *%rax* have the same referent as they both refer to the *%ax* location in memory. The registers `%rsp` and **%rbp** have unique distinct referents as they refer to the **%sp** and **%bp** memory locations which no other registers in the block refer to.

Figure 1(b) shows a semantically equivalent version of the block as all registers are renamed in a bijective manner that preserves which tokens share the same referent and that also preserves the view of each token. To generate this new block, `%rsp` was renamed to `%r8` and all other registers remained fixed. Since `%r8` is also a 64 bit register, this mapping preserves the view of each register. Since `%r8` occupies the `%8` location in memory, which is distinct from all other registers, `%r8` has its own unique referent. Thus as `%rsp`, which had a unique referent in the original block, is mapped to a new register which also has a unique referent in the renamed block, all registers which shared a unique referent still share a unique referent. Because the permutation preserves the semantics of the original block, llvm-mca outputs the same timing for the renamed block as the original block.

Figure 1(c) shows a version of the block with registers renamed in manner that is not semantically equivalent, as it both breaks which registers share the same referent and does not preserve

the view of each register. First, views are not preserved: **%rbp**, a 64 bit register, is renamed to **%ebp**, a 32 bit register, thus changing the execution time of the instruction. Second, which tokens share the same referent is not preserved: **%rsp** is renamed to **%rax** on the first line while *%rax* and *%eax* are not renamed. This creates a new dependency in the renamed block as registers which originally referred to different locations in memory and thus had unique referents were mapped to registers which do share the same location in memory and now share the same referent. Because the permutation does not preserve the semantics of the original block, llvm-mva outputs a different timing for the renamed block compared to the original block.

**Renaming invariance in Transformers.** Renda et al. use a Transformer (Vaswani et al., 2017) as the surrogate architecture for llvm-mca. This Transformer model takes the basic block as input, and outputs a prediction of what llvm-mca would output on that basic block.

The base Transformer architecture is sensitive to register renaming. For example, a BERT model trained as a surrogate of llvm-mca generates a prediction of 1.5 cycles on the block in Figure 1(a). This same model generates a prediction of 2.3 cycles on the block in Figure 1(b), demonstrating that it is not invariant to the register renaming transformation. In contrast, the Renamer model that we present in Section 4 generates a prediction for 1.6 cycles for both blocks in Figures 1(a) and 1(b). We provide a more in depth analysis in Appendix A.1.

## 3 RENAMING INVARIANCE

In order to describe an architecture which is invariant to renaming, we need a formal definition of functions which are invariant to renaming. In this section, we formally define what it means for a function to be renaming invariant.

**Formalism.** Let $x \in X$ be an input token from the set of input tokens and let $[x_i] \in X^n$ be a length $n$ sequence of tokens.

Let $f : X^n \to \mathbb{R}$ be a function over a sequence of tokens.

Let $R$ be a set of *referents*. Referents represent the underlying set of entities that can be renamed. Each token is associated with a single referent by the *referent mapping* $r : X \to R$.

Let $V$ be a set of *views*. A view represents the semantic information about a token that is relevant to the function output. Each token is associated with a view by the *view mapping*: $v : X \to V$.

Let $\sigma : X \to X$ be a permutation (i.e., a bijection) over the input tokens. $\sigma$ is *referent-constrained* if there exists a permutation over referents $\phi : R \to R$ such that $\forall x \in X . r(\sigma(x)) = \phi(r(x))$. This means that all tokens which originally shared the same unique referent are permuted such that they still all share a unique referent. This can also be stated as all tokens which originally referred to the same entity still all share the same entity after being permuted. $\sigma$ is *view-constrained* if $\forall x \in X . v(x) = v(\sigma(x))$, meaning that all tokens are mapped to a token with the same semantic role in the function.

Furthermore, $\sigma$ is *semantics-preserving* if it is both referent-constrained and view-constrained. $\sigma^n : X^n \to X^n$ is an element-wise extension of $\sigma$ to sequences such that all tokens in the sequence which have the same identity are mapped to the same value. Then a sequence processing function $f$ is *renaming invariant* for a given set of referents and views if for all semantics-preserving permutations $\sigma^n$, $f([x_i]) = f(\sigma^n([x_i]))$.

## 4 RENAMER ARCHITECTURE

Renamer enforces the renaming invariance by modifying the first layer of the Transformer. Technically, for any semantics-preserving renaming, the output Renamer must be identical for the original and renamed input. Renamer must not only be renaming invariant, but must also preserve the representational capacity of a vanilla Transformer. The remainder of this section gives background on Transformer models, then describes our implementation.

## 4.1 Transformers Background

Renamer modifies the embeddings and self-attention mechanisms of Transformer networks. A Transformer takes a sequence of tokens $[x_i]$ as input. Given $[x_i]$, the Transformer computes a content embedding $C^{[x_i]} \in \mathbb{R}^{n \times d}$ token-wise as $C_j^{[x_i]} = C([x_i]_j)$, where $C : X \to \mathbb{R}^d$ is an embedding table. The identity of a token in $[x_i]$ is exposed to the Transformer through its content embedding. A positional embedding $P^{[x_i]} \in \mathbb{R}^{n \times d}$ is also computed token-wise as $P_j^{[x_i]} = P(j)$, where $P : \mathbb{N} \to \mathbb{R}^d$ is an arbitrary function. The only restriction on $P$ in this work is that $P$ assigns a unique embedding for each index in the sequence. The embedding for a sequence $[x_i]$ is $E^{[x_i]} = C^{[x_i]} + P^{[x_i]}$ where $+$ denotes element-wise addition. Self-attention is defined as $\text{softmax}(M(QK^T))V$. $Q \in \mathbb{R}^{n \times d_{attn}}$, $K \in \mathbb{R}^{n \times d_{attn}}$, and $V \in \mathbb{R}^{n \times d}$ are distinct linear projections of the embedding $E^{[x_i]}$. $M : \mathbb{R}^{n \times n} \to \mathbb{R}^{n \times n}$ is an attention mask operator computed from $[x_i]$ that sets the unnormalized attention score of token pairs which should not attend to each other to $-\infty$ such that they do not contribute to the normalized attention scores.

## 4.2 Renamer Architecture Modifications

**View anonymization.** So that all input tokens which share the same view share the same representation, the same content embedding is shared between all tokens that share the same view:

$$\forall x_j, x_k \in X . v(x_j) = v(x_k) \Rightarrow C(x_j) = C(x_k)$$

Then $\forall x_j \in [x_i] . C(x_j) = C(\sigma(x_j)) \Rightarrow C^{[x_i]} = C^{\sigma^n([x_i])}$. Thus $f([x_i]) = f(\sigma^n([x_i]))$, meaning that after view anonymization the Renamer is renaming invariant.

Concretely, we implement view anonymization by mapping each token in the input sequence to a token that represents its view.

**Referent binding.** View anonymization alone limits the class of functions the Transformer can represent. All elements in the same view share the same content embedding, so the network can't differentiate which tokens in the input do or do not have the same referent under the referent mapping.

So that all renaming invariant functions representable by a vanilla Transformer are still learnable by Renamer, the network must be able to differentiate inputs based on which tokens share the same referent and which do not. Tokens with different referents are disambiguated by first applying attention only between tokens with the same referent. Given an input sequence $[x_i]$ and a referent mapping $r$, we construct a *referent attention mask operator* $M^r$ as follows:

$$M_{j,k}^r(S) = \begin{cases} S_{j,k}, & r([x_i]_j) = r([x_i]_k) \\ -\infty, & \text{otherwise} \end{cases}$$

The representation after referent binding is then: $\text{softmax}(M^r(QK^T))V$. In words, the referent attention mask operator is defined such that attention is only computed between tokens which share the same referent. Applying the referent attention mask gives the network the capacity to represent functions which depend on the relations between tokens with the same referent. While the same content embedding is shared for all tokens in the same view, by assumption the positional embedding for each token is unique. Thus, in only attending to other registers in the same referent group, the representations for each token with the same referent can be a combination of unique embeddings.

## 5 Evaluation

We evaluate Renamer by learning a surrogate of llvm-mca, comparing against a suite of baselines.

## 5.1 Task

The task under study is to take an x86-64 basic block as input, and output a prediction of the timing that llvm-mca would output for this basic block.

**Register renaming invariance in llvm-mca.** We define the set of referents to be the set of abstract locations where register data is stored. Thus the set of referents for x86-64 is:

$$\{\text{AX,BX,CX,DX,SI,DI,SP,BP}\}\cup\{\text{R}n\,|\,n\in\{8,\cdots,16\}\}\cup\{\text{XMM}n,\text{YMM}n\,|\,n\in\{0,\cdots,7\}\}$$

The referent for a given register is the abstract location where that register data is stored: for example, referent mapping $r$ has $r(\%\text{rax})$=AX, $r(\%\text{eax})$=AX, $r(\%\text{r12w})$=R12. All other tokens have unique singleton referents.

The views are defined as the set of possible bit-widths: {8-bit, 16-bit, 32-bit, 64-bit, 128-bit, 256-bit}, combined with the set of register classes {general-purpose, floating-point, vector}. For example, view mapping $v$ has $v(\%\text{rax})$=(64-bit, general-purpose), $v(\%\text{eax})$=(32-bit, general-purpose), $v(\%\text{xmm0})$=(128-bit, vector). All other tokens have unique singleton views.

With the exception of instructions with *implicit operands*, those instructions which read from or write to specific hard-coded registers, llvm-mca is renaming invariant with these referents and views. Instructions with implicit operands break renaming invariance because a permutation cannot change implicit registers. Such instructions include `push`, `pop`, `mul`, `div`, and others.

**Dataset.** We evaluate the Renamer on the BHive dataset (Chen et al., 2019), which is a collection of x86-64 basic blocks from a variety of real-world programs. The full BHive dataset consists of 287639 basic blocks. We remove inputs which have implicit operands from this dataset. We then perform a random 70/10/20 split on the original dataset, resulting in a dataset with 185,773 blocks in the training set, 26,107 blocks in the validation set, and 52,278 blocks in the test set.

## 5.2 MODELS

For all experiments, we use a BERT Transformer (Devlin et al., 2019) as the backbone of our architecture. We evaluate on BERT-Tiny, BERT-Mini, and BERT-Small (Turc et al., 2019).

**Vanilla Transformer.** The vanilla Transformer, which serves as a backbone for all other models, is an encoder-only BERT (Devlin et al., 2019), with absolute positional encodings.

**Augmented Transformer.** In addition to the vanilla Transformer, we validate against an augmented training baseline. The architecture for the augmented baseline is identical to a vanilla BERT, the only difference is that for each batch during training a semantics-preserving register permutation is applied to the registers of the basic blocks. The specifications of such permutations are provided in Section 2. While the resulting model is not guaranteed to be invariant to variable renaming, this training paradigm removes any bias towards specific registers in the dataset.

**Canonicalized Transformer.** We also evaluate a Transformer model which *canonicalizes* basic blocks before using them as input to the vanilla Transformer. Canonicalization takes each basic block and maps it to a unique *canonical* basic block; canonicalization maps all semantics-preserving transformation of a basic block to the same canonical block. This preprocessing scheme ensures that the Transformer is invariant to variable renaming.

**Renamer.** Renamer models have the same architecture as the vanilla model, except for the first encoder block which employs view masking and referent binding as described in Section 4. View masking and referent binding do not change the number of parameters of the Transformer.

## 5.3 EVALUATION METHODOLOGY

**System.** We evaluate using Pytorch-1.2.0 (Paszke et al., 2019), HuggingFace 4.17.0 (Wolf et al., 2020). Training is performed using an NVIDIA Tesla-V100. Each reported metric is the mean and the standard error of that metric across five trials with different random seeds.

**Hyper-parameters.** Across all models, we use the AdamW optimizer with a $\beta_1$, $\beta_2$ of 0.9 and 0.999 respectively. The Tiny and Mini models use a learning-rate of 0.0003 while the small models use a learning rate of 0.0001. All models use a weight-decay of 0.01 and a dropout

Table 1: MAPE on original test set

| Model | Model size | | |
|---|---|---|---|
| | BERT-Tiny | BERT-Mini | BERT-Small |
| Vanilla | 3.30% ±0.16% | 1.13% ±0.17% | 1.25% ±0.43% |
| Augmented | 3.34% ±0.11% | 2.25% ±0.02% | 2.36% ±0.02% |
| Canonicalized | 3.03% ±0.33% | 0.96% ±0.06% | 0.76% ±0.04% |
| Renamer | **2.39% ±0.07%** | **0.85% ±0.07%** | **0.59% ±0.06%** |

rate of 0. We empirically determine the learning-rate, weight-decay, and dropout through a grid search over the hyper-parameters, selecting the hyper-parameter configuration which has the lowest validation error for the vanilla model. The hyper-parameters swept over are: learning-rate {0.0003, 0.0001, 0.00005, 0.00001}, weight-decay {0.0, 0.01}, and dropout {0.0, 0.1}. All models have a batch size of 64, max sequence length of 128, and are trained for 500 epochs following Renda et al. (2021).

**Objective.** Following Chen et al. (2019) the loss and error metric are identical and are defined as the mean absolute percentage error (MAPE): $\mathcal{L}_{\text{MAPE}} = \sum_{x,y \in D} \frac{|f(x) - y|}{y}$

## 5.4 Results

**Best performance.** In this section we evaluate the best performance of the vanilla, augmented, canonicalized, and Renamer models. This is defined as the test error of the epoch with the lowest validation error.

Table 1 shows the error of each model across BERT sizes. We find that across all model sizes, the Renamer model outperforms the vanilla, augmented, and canonicalized models on the original test set. Renamer has a relative decrease in error compared to the vanilla model of 27.58%, 24.79%, and 52.80% for the Tiny, Mini, and Small BERT variants respectively. Additionally, the Renamer has a decrease in error as compared to the augmented model of 28.44%, 62.22%, and 75.00% for the Tiny, Mini, and Small BERT variants respectively. Finally, Renamer has a relative decrease in error compared to the canonicalized model of 21.12%, 11.46%, and 23.07% for the Tiny, Mini, and Small BERT variants respectively.

Additionally, we find that as model size is increased, Renamer suffers less from diminishing returns as compared to the vanilla and augmented models. For the vanilla model, the relative decrease in error between the Tiny and Mini model is 66.76%, while error increases by 10.62% between the Mini and Small model. Likewise for the augmented model, the error decreases by 32.63% for Tiny to Mini and increases by 4.89% for Mini to Small. In contrast, the Renamer decreases in error between both size variants by 64.44% and 30.59%. While both the vanilla and augmented models have a relative increase in error between BERT-Mini and BERT-Small, Renamer has a significant decrease in error.

**Renamed registers.** In this section we evaluate the error of the best performing vanilla, augmented, canonicalized, and Renamer models on a semantics-preserving register renamed version of the test set.

For each basic block in the original test set, a random semantics-preserving permutation is applied to the registers of the basic block. The checkpoint of the model with the lowest error on the unperturbed validation set is selected and then evaluated on the renamed test set.

Table 2 shows the error of each model across BERT sizes on the permuted version of the test set. The performance of the vanilla model is significantly affected by permuting the registers. Compared to the original, unperturbed test set, the error of the vanilla model increases by 59.39%, 144.25%, and 131.20% for the Tiny, Mini, and Small BERT variants respectively. This increase in error further empirically demonstrates the renaming sensitivity of the vanilla network architecture.

Table 2: MAPE on test set with renamed registers

| Model | Model size | | |
| --- | --- | --- | --- |
| | BERT-Tiny | BERT-Mini | BERT-Small |
| Vanilla | 5.26% ±0.37% | 2.76% ±0.42% | 2.89% ±0.52% |
| Augmented | 3.44% ±0.11% | 2.55% ±0.02% | 2.36% ±0.03% |
| Canonicalized | 3.03% ±0.33% | 0.96% ±0.06% | 0.76% ±0.04% |
| Renamer | **2.39% ±0.07%** | **0.85% ±0.07%** | **0.59% ±0.06%** |

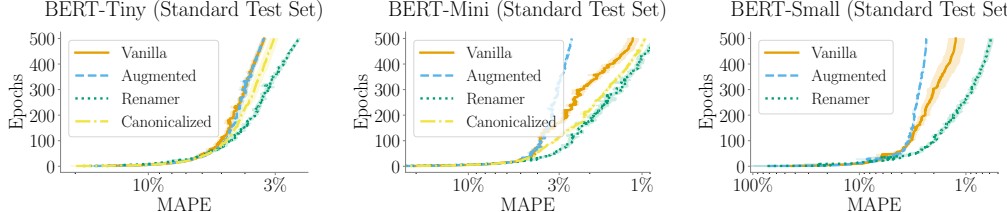

Figure 2: Model efficiency on the original (unperturbed) test set. Each point on a curve corresponds to the number of epochs needed to reach a given test error for the specified architecture.

In contrast to the vanilla model, Renamer is provably invariant to register perturbations. Accordingly, on the renamed test set, Renamer outperforms the vanilla model by 54.56%, 69.20%, and 79.58% for the Tiny, Mini, and Small BERT variants respectively.

While the augmented training model is also less sensitive to register permutations and the canonicalized model is also guaranteed to be invariant, Renamer still outperforms both the augmented training and canonicalized models on the renamed test set.

**Training efficiency.** In this section we evaluate the efficiency with which the vanilla, augmented, canonicalized, and Renamer models achieve various test accuracies. The time to process an input is the same across all models as they all have the same number of parameters, and Renamer only leverages components of the Transformer architecture (i.e. embedding table, attention mask) which the vanilla, augmented, and canonicalized architectures also employ. Since the time to train for an epoch is the same for all architectures, we directly compare the number of epochs needed to achieve comparable test error.

Figure 2 plots efficiency curves for the various model sizes on the BHive dataset. Each line shows the epochs required (on the y axis) to reach a given test error (on the x axis) for a given modeling approach. A more efficient model requires fewer epochs to reach a given target test error (i.e., lower is better).

Across all model sizes, Renamer achieves the same performance with significantly fewer training steps than the vanilla, augmented, and canonicalized models. This speedup is reflected in the gap between the MAPE versus epochs of training curves for the vanilla, augmented, canonicalized, and Renamer models.

We evaluate the speedup quantitatively by comparing the minimum number of epochs required to achieve the same best test error. Renamer reaches the best error of the vanilla model with 213, 123, and 290 fewer epochs for the Tiny, Mini, and Small architectures respectively, which corresponds to a relative decrease in steps to achieve the same performance of 42.77, 24.75, and 59.06%. As compared to the augmented model, Renamer requires 214, 403, and 419 fewer epochs for the Tiny, Mini, and Small variants respectively, which corresponds to a relative decrease in steps to achieve the same performance of 42.89, 81.25, and 83.80%. Finally, Renamer achieves the same error as the canonicalized model with 155 and 59 fewer epochs for the Tiny and Mini variants respectively, which corresponds to a relative decrease in training steps of 31.00 and 11.82%.

## 6 DISCUSSION

In this section we analyze the results and limitations in more detail.

**Source of improvement.** We have demonstrated that the Renamer achieves lower error faster than both the vanilla network and the augmented baseline. We hypothesize that this is because the Renamer only represents the subset of functions that are renaming invariant. Because llvm-mca itself is renaming invariant, this leads to a smaller search space for SGD, along with the guarantee that all solutions match llvm-mca's renaming invariance.

**Spurious correlation.** Though we find that on the BHive dataset the invariant model improves training and reduces test the error, this may not happen on all datasets. Specifically, we hypothesize that datasets with a spurious correlation between referents and labels even when the underlying function is renaming invariant. For instance, a version of BHive that only used the `%rax` register when the label is less than 100 cycles would have this property, even though the function being modeled is still renaming invariant. Though the in-distribution error of such a dataset may suffer with the invariant model, the transformed register evaluation performed in Section 5 would still result in the invariant model having better error.

## 7 RELATED WORK

**Anonymization and canonicalization.** Anonymizing and the canonicalization of training data is an area of much focus in fields ranging from ethical AI to privacy-preserving AI. In an effort to reduce gender and region bias in gendered pronoun resolution Liu (2019) mask individual names by drawing from a set of canonical names. Similarly, for debiasing and preserving privacy in clinical ml, de-identification of data is a prevalent technique (Dernoncourt et al., 2016; Liu et al., 2017; Johnson et al., 2020; Minot et al., 2022). Most of the work; however, is focused on the process of automatic de-identification and not on the result of training on de-identified data. Minot et al. (2022) investigate the result of training on a canonicalized version of medical records. While canonicalization and de-identification provide the benefit of reducing the bias of the model, they suffer from the fact that not every canonical representation of the same entity is guaranteed to have the same representation and that inputs which have more entities than the number of canonical representations trained can't be represented. Furthermore, training on de-identified data is often associated with a degradation to network performance.

**Transformer invariances.** A wide variety of invariances and equivariances have been encoded in the architecture of Transformer networks. Lee et al. (2019) introduce Set Transformer, a Transformer invariant to permutations of the ordering of the input sequence. Fuchs et al. (2020) suggest SE (3), a Transformer architecture which is equivariant to 3D translations and rotations, which they evaluate on a variety of domains ranging from n-body simulations to point-cloud object classification. Translation invariance has also been exploited in the context of natural language tasks (Su et al., 2021; Wennberg and Henter, 2021). While these works enforce invariances, they all deal with spatial or positional invariances. To our knowledge, there is limited prior work on encoding invariances regarding the content of individual tokens.

## 8 CONCLUSION

This work formalizes the concept of renaming invariance, and proposes the first Transformer architecture that is invariant to semantics-preserving variable renaming through the use of view anonymization by shared content embeddings and a novel restricted attention map that binds referents and restores the network's representational capacity. Our results demonstrate that on the BHive dataset, the invariant architecture substantially improves the best performance of the network on the clean test set, a register renamed test set, and delivers significant training speed-ups. This work is an important first step to designing renaming invariant neural networks, which is a ubiquitous and understudied invariance.

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

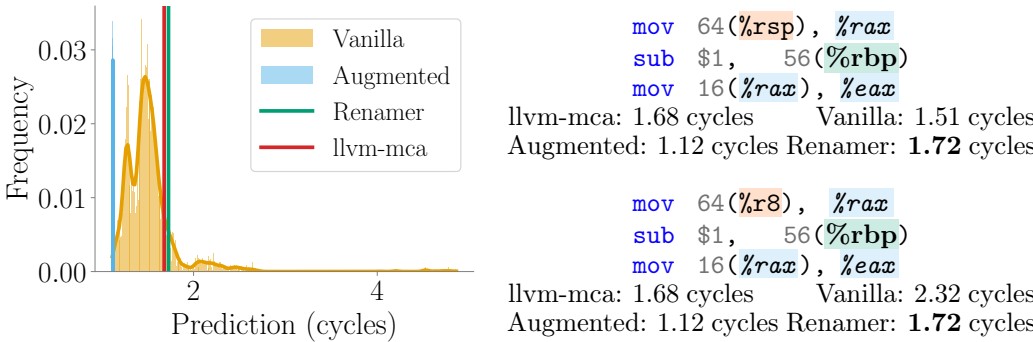

Figure 3: The range of generated predictions for renamings of the basic block in Figure 1(a).

## A   APPENDIX

### A.1   RENAMING SENSITIVITY CASE STUDY

We now present a more in-depth case study of the basic block presented in Figure 1.

Figure 3 presents a case study of each model's predictions on the basic block presented in Figure 1(a). The figure on the left is a histogram and corresponding density plot of predictions on semantically equivalent renamings of the basic block. To generate this plot, we uniformly sample 100,000 valid semantics-preserving register permutations $\sigma$, apply the permutation to the original block, then evaluate each model on the permuted block. The models under study are single trials of the best-performing BERT-Tiny models. The ground-truth timing for this basic block as output by llvm-mca is 1.68 cycles.

The vanilla model generates a range of predictions for permutations for this block, ranging from 1.11 cycles to 4.86 cycles. The predictions generated by the vanilla model are multimodal, though there are no clear indicators for which mode a given block will induce. The augmented model generates a significantly smaller range of predictions – though there is some variation on the order of one thousandth of a cycle, the predictions are essentially constant at 1.12 cycles. By construction, Renamer generates constant predictions for this basic block of 1.72 cycles.

### A.2   SYMBOLIC ALGEBRA CASE STUDY

In this section we study how the Renamer architecture can be extended to other tasks. Specifically, we apply Renamer to the task of determining whether one equation is the partial derivative of another.

**Dataset.**   We evaluate on a variant of the Backward dataset presented by Lample and Charton (2020). The dataset is a collection of 300,000 pairs of expressions. Each expression contains up to three different variables $(x,y,z)$ and up to 3 different symbolic coefficients $(a0,a1,a2)$. The task is to determine whether the first expression is the partial derivative of the second with respect to $x$. The expressions are in prefix notation. For example, the network should predict true for the pair of expressions `a0, mul a0 x`, since the first expression is the partial derivative of the second with respect to $x$. The network should predict false for the pair of expressions `a0, mul a0 y`, since the first expression is not the partial derivative of the second with respect to $x$.

**Variable renaming in symbolic algebra task.**   We define the set of referents to be the names of the variables. Thus the set of referents is $\{a0,a1,a2,x,y,z\}$.

We define a view as a group of variables which have the same functional purpose. Thus all coefficients share the same view $a$. Since $x$ is the variable we differentiate with respect to, it has a singleton view $x$. Finally, as $y$ and $z$ are both variables that are inputs to functions but are not differentiated, they share a view $s$. Thus the set of views is $\{a,x,s\}$.

Table 3: Accuracy of symbolic algebra case study.

| Test set | Model type | | |
|---|---|---|---|
| | Vanilla | Canonicalized | Renamer |
| Original | 99.21% ±0.13% | **99.29% ±0.14%** | 99.20% ±0.06% |
| Augmented | 95.32% ±0.69% | 95.02% ±0.56% | **97.62% ±0.16%** |

**Models.** For all experiments we use a BERT-Small (Devlin et al., 2019). We compare the performance of Renamer against a vanilla and canonicalized baseline.

**Results.** In Table 3 we evaluate all models on both the original test set and an augmented version of the test set. To build the augmented test set, we introduce a new coefficient $a3$ to each equation. We find that all models perform similarly on the original test set. While the canonicalized model has the highest average accuracy, it only outperforms Renamer by 0.09% accuracy. While the performance of all models decreases on the augmented test set, we find that on the augmented test set Renamer significantly outperforms all other models. The vanilla model achieves an accuracy of 95.32% on the augmented test set while Renamer reaches an accuracy of 97.62% which is an improvement in accuracy of 2.30%. Similarly, the canonicalized model achieves an accuracy of 95.02% which means Renamer has an improvement in accuracy of 2.60%.

