# OpenReview forum: "Renamer: A Transformer Architecture In-variant to Variable Renaming"
_ICLR.cc/2023/Conference — Submitted to ICLR 2023_

### Official Review · Reviewer_1jmm · 2022-10-24

**Confidence:** 5
**Correctness:** 3
**Technical Novelty And Significance:** 2
**Empirical Novelty And Significance:** 2
**Recommendation:** 5

**Clarity, Quality, Novelty And Reproducibility:**

The description of Renamer implementation is not clear enough.

The proposed approach is not technically very novel. The separation of view and referent seems specific to the task in their evaluation, and does not add much to the variable renaming invariance that has been discussed in prior works.

The authors did not provide the code and data for their experiments. As the technical details are not clear, it could be hard to reproduce their results.

**Strength And Weaknesses:**

Strengths:

In general, renaming invariance is an important desired property of neural networks for code, and the authors present a domain where achieving renaming invariance improves the performance over baseline architectures.

Weaknesses:

One key issue of this work is that the empirical study is not solid. First, NA in Table 1 and Table 2 is not well justified. Training with canonicalization does not add more computational costs compared to other methods, while it achieves much better results than other baselines. Therefore, even if the authors do not have enough computational resources to evaluate larger BERT models, which should further improve the results, at least the authors should complete the evaluation of smaller BERT models.

Furthermore, there is no comparison to SOTA models to justify the significance of the results. To meet their assumptions of register renaming invariance, the authors only train and evaluate on a subset of the original BHive benchmark, where they remove inputs with implicit operands. This makes it difficult to directly compare numbers in this paper to prior works that evaluate on this benchmark. Thus, the authors should evaluate the SOTA models on their subset and present the results.

Meanwhile, although the paper starts with a general definition of renaming invariance, the separation of view and referent seems specific to the task in their evaluation, and does not add much to the variable renaming invariance that has been discussed in prior works. In particular, there is no analysis on why and how Renamer outperforms canonicalization.

Also, the description of Renamer architecture modification is unclear. My understanding is that view anonymization is similar in spirit to canonicalization, while referent binding adds additional constraints on the Transformer attention. In this case, I think Renamer does not preserve the expressive power of Transformer with canonicalization. More explanation about the implementation of modifications is helpful.

**Summary Of The Paper:**

This paper proposes Renamer, which builds upon the Transformer architecture to achieve variable renaming variance. Their approach involves two parts: view anonymization and referent binding, and they focus on representing x86-64 blocks in their implementation. Specifically, view anonymization maps all registers with the same bitwidth to the same token for input embedding, and referent binding adds an attention mask so that the attention is computed only among registers referring to the same location in memory. They evaluate their approach on BHive benchmark for throughput prediction, and add their Renamer upon BERT-Tiny, BERT-Mini, and BERT-Small. Empirical results show that their Renamer variants outperform the baseline BERT architectures on both the original test set and a variant of the test set with renamed registers.

**Summary Of The Review:**

Proposing approaches to achieve renaming invariance while preserving the overall task performance is a good research topic. However, the evaluation in this work is not solid, the description of their approach is unclear, and the implementation of Renamer seems specific to the task in their evaluation. Therefore, I recommend a rejection.

---------------
I thank the authors for the response and paper revision. The added experiments on Backward dataset help me better understand the approach. However, I still think the technical novelty is limited, and the applicability of the approach is a bit narrow. Therefore, I slightly improve my score to 5.

---

> ### Author Response · Authors · 2022-11-19
> **Response to Reviewer 1jmm**
>
> > Therefore, even if the authors do not have enough computational resources to evaluate larger BERT models, which should further improve the results, at least the authors should complete the evaluation of smaller BERT models.
>
> Please see our [general response](https://openreview.net/forum?id=7hYCGFacpz&noteId=woo_pw3nky) for a discussion of the size of the models evaluated, and of the canonicalized model on BERT-Small.
>
> > … the authors only train and evaluate on a subset of the original BHive benchmark … the authors should evaluate the SOTA models on their subset and present the results.
>
> The state-of-the-art published model on this task is the Ithemal model described in [1], a hierarchical LSTM model with hidden size 256. The [published version of the Ithemal model](https://github.com/ithemal/Ithemal-models/tree/master/paper/haswell) achieves an error of 11.9% on our subset of the dataset. Training Ithemal from scratch on this dataset yields a model with an error of 5.9% ± 0.08%, higher than that of the smallest baseline we evaluate, BERT-Tiny. We will include this comparison in the final version of the paper.
>
> -----
> > Also, the description of Renamer architecture modification is unclear. My understanding is that view anonymization is similar in spirit to canonicalization, while referent binding adds additional constraints on the Transformer attention… More explanation about the implementation of modifications is helpful
>
> Section 4 discusses the implementation of the Renamer:
> > [Section 4:] Concretely, we implement view anonymization by mapping each token in the input sequence to a token that represents its view.
> > …
> > the referent attention mask operator is defined such that attention is only computed between tokens which share the same referent.
>
> We would be happy to make any proposed changes that further increase the clarity of the paper.
>
> -----
>
> > …I think Renamer does not preserve the expressive power of Transformer with canonicalization.
>
> We believe there is a misunderstanding about our meaning of expressive power. Recall that as stated in Sections 1 and 4, the Renamer applies the restricted attention map only in the first layer. A more formal version of the statement in the paper is, given the same number of unrestricted attention layers, the Renamer can express any renaming invariant function that a Transformer with canonicalization can (assuming unique positional embeddings). We stand by this statement. The reviewer correctly notes that the renamer has a restricted first layer compared to a Transformer with canonicalization, meaning that in our evaluation (where we use the same total number of layers), the Renamer does have less capacity (as measured by the number of FLOPs).
>
> We will include this more nuanced discussion in the final version of the paper.
>
> > The separation of view and referent seems specific to the task in their evaluation…
>
> Nontrivial views and referents arise in any task in which not all variables can be renamed with each other. For example, this is the case in symbolic mathematics, where symbolic coefficients are treated differently than variables [2, Section 3.4]; in typed languages, where variables of different types may require different interpretations; and in natural language, where for example names and pronouns may refer to the same underlying entity (i.e., share a referent) but have different grammatical parts of speech (i.e., have different views).
>
> > The proposed approach is not technically very novel. The separation of view and referent … does not add much to the variable renaming invariance that has been discussed in prior works.
>
> We are not aware of prior work that has formalized the problem of variable renaming invariance, nor of any that has developed renaming invariant architectures. The formalization of views and referents is key to these two novel contributions.
>
> > The authors did not provide the code and data for their experiments.
>
> We will provide the full code and data for the final version of the paper.
>
> ## References
>
> [1] Charith Mendis, Alex Renda, Saman Amarasinghe, Michael Carbin. Ithemal: Accurate, Portable and Fast Basic Block Throughput Estimation using Deep Neural Networks. ICML, 2019.
>
> [2] Guillaume Lample, François Charton. Deep Learning for Symbolic Mathematics. ICLR, 2020.

---

### Official Review · Reviewer_KJQk · 2022-10-25

**Confidence:** 4
**Correctness:** 4
**Technical Novelty And Significance:** 3
**Empirical Novelty And Significance:** 3
**Recommendation:** 6

**Clarity, Quality, Novelty And Reproducibility:**

The paper is clear. It should be reproducible based on the method description and training setup. To my knowledge the work original. Transformers with invariances exist. There are invariances similar in spirit in natural language but their definition is unique enough to be novel.

**Strength And Weaknesses:**

The paper develops good intuition with the application to formal languages and concrete examples. Experimental results are very strong on standard task.

The main weakness is that they only work with one dataset, so it's hard to know how much the technique generalizes. Since they mention applications to mathematics or synthetic grammars, it would have been nice to see such experiments.

**Summary Of The Paper:**

The paper introduces the concept of renaming invariance and develops a model with this property. They carry out experiments and demonstrate that their model performs better.

**Summary Of The Review:**

The paper is well-timed to be interesting given the interest in apply machine learning to code and mathematics.

---

> ### Author Response · Authors · 2022-11-19
> **Response to reviewer KJQk**
>
> > The main weakness is that they only work with one dataset, so it's hard to know how much the technique generalizes
>
> Please see our [general response](https://openreview.net/forum?id=7hYCGFacpz&noteId=woo_pw3nky) for a discussion of other tasks.

---

### Official Review · Reviewer_uTBf · 2022-10-27

**Confidence:** 4
**Correctness:** 4
**Technical Novelty And Significance:** 3
**Empirical Novelty And Significance:** 3
**Recommendation:** 6

**Clarity, Quality, Novelty And Reproducibility:**

The work is communicated well.

The technique is novel, but potentially narrow.

**Strength And Weaknesses:**

The paper is well presented, both the task and architecture changes are well motivated and described clearly. The specific task studied is interesting and the proposed changes might also benefit other tasks.

The main weakness of the paper is its narrow scope.

A single dataset is used for evaluation. Further, inputs which contain implicit operands are removed from the evaluation. These operations read to / write from hardcoded registers, so these are not invariant to renaming. Could they have been kept in the dataset but the hardcoded variables handled differently from those that can be renamed?

The explanation of view is closely tied to this task. Considering other tasks that are invariant to variable renaming, what is the equivalent concept and what are the implications for the proposed method? For example, if evaluating C code, variable types provide semantic information which can be separated from information about the referent. What about working with a dynamically typed language? Or similarly, with the masked attention stage, what is the effect of having a variable reassigned during a code block so that its referent changes?

Overall, the paper neither seriously discusses nor tests whether the proposed method is applicable for any other tasks which are invariant to renaming of variables. To make the work more impactful it should be clarified whether and how the method would need to be modified for other tasks, preferably with demonstration of its effectiveness through additional evaluations.

Only small models are evaluated, and benefits observed in small models do not always remain for larger models. This is a minor weakness. Being able to achieve good results using small models is valuable in itself; with evaluations on larger models we might also see how much is gained by architectural changes vs more compute.

Questions:

It’s interesting that the augmented model shows only a very small drop in performance when tested on renamed inputs, but overall performs less well than the model trained without data augmentation. Did you try training for longer with more variants? I’d be interested to know the limits of data augmentation to improve robustness to renaming in this case.

Why did the canonicalized model take longer to train? My understanding is that the inputs were first modified to a standard version, then the model trained on these. It’s not immediately obvious to me why this should take much longer to train than the basic setup. I may have misunderstood something here.


**Summary Of The Paper:**

This work presents a modified Transformer (Renamer) which is invariant under renaming of variables. The design is tested on a task to predict CPU clock cycles to execute blocks of assembly language. The proposed model is compared against an unmodified Transformer which is optionally also trained on inputs with renamed variables (augmented), or adds a preprocessing step to fix variable names for the input (canonicalized). Renamer reduces the error for the studied task compared to all three baseline models.


**Summary Of The Review:**

I would recommend to accept this paper for publication. The modification to the standard Transformer architecture is simple but effective.

---

> ### Author Response · Authors · 2022-11-19
> **Response to Reviewer uTBf**
>
> > Could [implicit operands] have been kept in the dataset but the hardcoded variables handled differently from those that can be renamed?
>
> Assigning unique views to hardcoded variables (so that they can be distinguished from rename-able variables) is possible within our formalism and would successfully solve this challenge. However, this approach would distinguish such registers even in basic blocks without implicit operands, thereby not expressing the full renaming invariance present in llvm-mca. We chose to instead remove these from the dataset to demonstrate the full renaming invariance, and also to simplify presentation.
>
> > Considering other tasks that are invariant to variable renaming, what is the equivalent concept and what are the implications for the proposed method? For example, if evaluating C code, variable types provide semantic information which can be separated from information about the referent. What about working with a dynamically typed language?
>
> The specific choices of views depends on the exact task under consideration, but the general intuition (as stated in the introduction) is that the view is the semantic information about the token that is salient to the task. As you say, in C code this may be the type. If types are unavailable as in a dynamic language, all variables may have the same view, as long as this is sufficient to encode the task.
>
> > … with the masked attention stage, what is the effect of having a variable reassigned during a code block so that its referent changes?
>
> In our formalism and implementation, referents are a property of tokens, not of variables. So, even in the presence of a statement like `x = y`, `x` and `y` would maintain separate referents. Though a more fine-grained analysis may detect that after such an assignment, `x` changes to become an alias of `y`, incorporating such an analysis into the Renamer is a significant additional research challenge beyond the scope of this paper.
>
> > the paper neither seriously discusses nor tests whether the proposed method is applicable for any other tasks which are invariant to renaming of variables.
>
> Please see our [general response](https://openreview.net/forum?id=7hYCGFacpz&noteId=woo_pw3nky) for a discussion of other tasks.
>
> > Only small models are evaluated…
>
> Please see our [general response](https://openreview.net/forum?id=7hYCGFacpz&noteId=woo_pw3nky) for a discussion of the scale of experiments.
>
> > … and benefits observed in small models do not always remain for larger models.
>
> Across all model sizes, the Renamer outperforms all baselines. As measured by the percentage decrease in error, the extent to which the Renamer outperforms baselines increases as size increases in all settings except for the canonicalized model on BERT-Mini.
>
> Increase in error of baseline over rename on standard test set (higher means Renamer is relatively performing better):
>
> ```
> Model size    Vanilla    Augmented    Renamer    Canonicalized
> ------------  ---------  -----------  ---------  ---------------
> BERT-Tiny     1.40x      1.40x        1.00x      1.27x
> BERT-Mini     3.01x      1.34x        1.00x      1.14x
> BERT-Small    4.00x      2.12x        1.00x      1.31x
> ```
>
> Increase in error of baseline over rename on augmented test set (higher means Renamer is relatively performing better):
> ```
> Model size    Vanilla    Augmented    Renamer    Canonicalized
> ------------  ---------  -----------  ---------  ---------------
> BERT-Tiny     1.40x      2.23x        1.00x      1.27x
> BERT-Mini     3.01x      3.26x        1.00x      1.14x
> BERT-Small    4.00x      4.92x        1.00x      1.31x
> ```
>
> > Did you try training for longer with more variants? I’d be interested to know the limits of data augmentation to improve robustness to renaming in this case.
>
> We agree that understanding the limits of data augmentation versus explicit architecture invariance is an interesting question. While we did not try training for longer than reported in the paper, the current training time is already long (500 epochs). Also as can be seen in Figure 2, the performance of the augmented models plateaus more significantly as compared to the other architectures suggesting fewer gains from longer training times.
>
> > Why did the canonicalized model take longer to train?
>
> Please see our [general response](https://openreview.net/forum?id=7hYCGFacpz&noteId=woo_pw3nky) for a discussion of the canonicalized model on BERT-Small.

---

### Author Response · Authors · 2022-11-19
**Response to all reviewers**

Thanks to all reviewers for their helpful feedback.

## Symbolic integration task

> [Reviewer uTBf:] Overall, the paper neither seriously discusses nor tests whether the proposed method is applicable for any other tasks which are invariant to renaming of variables...

> [Reviewer KJQk:] The main weakness is that they only work with one dataset...

> [Reviewer 1jmm:] the separation of view and referent seems specific to the task in their evaluation...

We have extended our evaluation of Renamer to a symbolic algebra task [1]. Appendix A.2 in the revised paper draft presents this experiment.

We evaluate a variant of the Backward dataset in [1], taking as input a pair of expressions and predicting whether or not one expression is the derivative of the other. We evaluate the original, canonicalized, and Renamer using a BERT-Small model. Full methodological details, including the set of views and referents for this task, are presented in Appendix A.2.

On the original test set, the vanilla, canonicalized, and Renamer models all have comparable performance, with the canonical model outperforming Renamer by 0.09% accuracy. We also evaluate the models with an augmented version of the test set which uses more coefficients than were observed at training time. On this augmented test set, Renamer outperforms the vanilla model by 2.30% accuracy and outperforms the canonicalized model by 2.60% accuracy. A full analysis of these results is presented in our revised paper (section A.2).

## Scale of models in experiments (Reviewers uTBf and 1jmm)
> [Reviewer uTBf:] Only small models are evaluated…

> [Reviewer 1jmm:] …even if the authors do not have enough computational resources to evaluate larger BERT models, which should further improve the results…

Though we agree that evaluating on the largest possible models is always the ideal, we would like to clarify the scale of the experiments already in the paper. The BERT-Small model we evaluate has 28,763,648 parameters, more than that of an ImageNet ResNet-50, and takes 90-100 GPU-hours to train per trial.

## Canonicalized BERT-Small

> [Reviewer uTBf:] Why did the canonicalized model take longer to train?...

> [Reviewer 1jmm:] Training with canonicalization does not add more computational costs compared to other methods…

Indeed, the canonicalized model does not take longer to train, but due to the time it takes to train (~4 days) we were not able to include it in the original submission. Since submission, we have completed the experiments for BERT-Small on the canonicalized dataset, and have included these results in the revision. On both the original test set (Table 1) and the perturbed test set (Table 2), this model has an error of 0.77%, which is 31% higher than that of the Renamer.

## References

[1] Guillaume Lample, François Charton. Deep Learning for Symbolic Mathematics. ICLR, 2020. https://arxiv.org/abs/1912.01412

---

### Decision · Program_Chairs · 2023-01-20

**Decision:**

Reject

**Justification For Why Not Higher Score:**

Limited scope of the problem and results.

**Justification For Why Not Lower Score:**

N/A

**Metareview: Summary, Strengths And Weaknesses:**

All reviewers find the paper to be well written - both the problem statement and the proposed solution are presented well. Reviewers mainly are worried about the narrow scope of the paper, evaluation on only one dataset and non-standard evaluation by removing datapoints with implicit operands. While authors added experiment results on another dataset during response, reviewers remain unconvinced about the general usefulness of the proposed approach. Hence I recommend rejection.